# Myeloid *TM6SF2* Deficiency Inhibits Atherosclerosis

**DOI:** 10.3390/cells11182877

**Published:** 2022-09-15

**Authors:** Wenzhen Zhu, Wenying Liang, Haocheng Lu, Lin Chang, Jifeng Zhang, Y. Eugene Chen, Yanhong Guo

**Affiliations:** 1Department of Internal Medicine, Frankel Cardiovascular Center, University of Michigan, Ann Arbor, MI 48109, USA; 2Department of Pharmacology, Southern University of Science and Technology, Shenzhen 518055, China

**Keywords:** *TM6SF2*, macrophage, cholesterol, atherosclerosis, ER stress, inflammation

## Abstract

Genetic variants in transmembrane 6 superfamily member 2 (*TM6SF2*), such as E167K, are associated with atherosclerotic cardiovascular disease (ASCVD). Chronic inflammation and lipid-laden macrophage foam cell formation are the central pathogeneses in the development of atherosclerosis. This study was undertaken to illustrate the biological function of *TM6SF2* in macrophages and its role during atherosclerosis development. We generated myeloid cell-specific *Tm6sf2* knockout mice on ApoE-deficient background (LysM Cre+/*Tm6sf2*^fl/fl^/*ApoE*^−/−^, TM6 mKO) with littermate LysM Cre−/*Tm6sf2*^fl/fl^/*ApoE*^−/−^ (Control) mice as controls. Mice were fed a Western diet for 12 weeks to induce atherosclerosis. Myeloid *Tm6sf2* deficiency inhibited atherosclerosis and decreased foam cells in the plaques without changing the plasma lipid profile. RNA sequencing of bone marrow-derived macrophages (BMDMs) from TM6 mKO mice demonstrated the downregulation of genes associated with inflammation, cholesterol uptake, and endoplasmic reticulum (ER) stress. *TM6SF2* was upregulated by oxidized low-density lipoprotein (oxLDL) in macrophages. Silencing *TM6SF2* in THP-1-derived macrophages and *Tm6sf2* deficiency in BMDMs reduced inflammatory responses and ER stress and attenuated cholesterol uptake and foam cell formation, while the overexpression of *TM6SF2* showed opposite effects. In conclusion, myeloid *TM6SF2* deficiency inhibits atherosclerosis development and is a potential therapeutic target for the treatment of atherogenesis.

## 1. Introduction

Atherosclerosis is a major cause of cardiovascular disease (CVD) and has been associated with increased levels of total cholesterol (TC), low-density lipoprotein cholesterol (LDL-C), and triglycerides (TG) and decreased high-density lipoprotein cholesterol (HDL-C) levels [1,2]. Atherosclerosis is a chronic inflammatory vascular disease in which the infiltration of macrophages and other immune cells and their interaction with arterial tissue play an important role in atherosclerosis progression [3]. Monocyte-derived macrophage uptake cholesterol resulting in foam cell formation and polarization to proinflammatory macrophages, contributing to the atherosclerotic plaque initiation and progression [4,5].

Transmembrane 6 superfamily member 2 (*TM6SF2*) was first identified in the EUROIMAGE consortium sequencing project in 2000 [6]. The human *TM6SF2* gene is located at chromosome 19p12 and is highly conserved in Homo sapiens and Mus musculus. Exome-wide and genome-wide association studies have identified that a variant in the *TM6SF2* gene (rs58542926, encoding p.Glu167Lys, E167K) is associated with reduced myocardial infarction risk and decreased plasma LDL-C levels but increased liver fat levels [7,8,9], demonstrating the paradoxical effect of *TM6SF2* variant in cardiovascular and liver diseases. The E167K mutation in *TM6SF2* yields a misfolded protein, causing accelerated protein degradation and reduced protein levels in hepatocytes [7]. TM6SF2 facilitates cholesterol synthesis and very-low-density lipoprotein (VLDL) assembly, and knockdown of *TM6SF2* reduces TG synthesis in human hepatocytes [10,11,12]. Consistent with human genetic findings, liver-specific *Tm6sf2*-deficient mice show significantly reduced TC and TG levels and VLDL particle size [10,11] while exhibiting spontaneous hepatic steatosis, fibrosis, and accelerated development of hepatocellular cancer [13]. Better understanding of the biological functions of *TM6SF2* in cells affected by atherosclerosis will elucidate the effects of *TM6SF2* in atherosclerosis and liver diseases, which may aid in developing new approaches for treating cardiovascular diseases.

Considering that inflammatory responses and lipid accumulation in macrophages are characteristic features of atherosclerotic plaques, in the current study, we sought to determine the contribution of macrophage *TM6SF2* to atherosclerosis. We found that TM6SF2 deficiency decreased inflammatory responses, inhibited foam cell formation, and reduced ER stress in macrophages. We demonstrated that the genetic ablation of *Tm6sf2* in myeloid cells of the *ApoE*-deficient (*ApoE*^−/−^) mice significantly attenuated atherosclerotic plaque formation without affecting circulating TC and TG levels. Our findings reveal that the suppression of macrophage TM6SF2 may become an appropriate therapeutic strategy for atherosclerosis treatment.

## 2. Materials and Methods

### 2.1. Animal Procedures

All animal experiments in this study were approved by the Institutional Animal Care and Use Committee (IACUC) and met the animal care standards and use established by the Animal Welfare Act and the NIH Guide for the Care and Use of Laboratory Animals. *Tm6sf2* floxed embryonic stem (ES) cell clone (EPD0097_3_E02, ES cell strain: C57BL/6N) were obtained from KOMP Repository at UC Davis. The exon 2 of the *Tm6sf2* gene was flanked by loxP sites. The ES clone was microinjected into blastocyst stage embryos to generate chimeric mice. The frt-flanked selection marker was removed by breeding with FlpO mice [14], yielding the floxed *Tm6sf2* mice (*Tm6sf2*^fl/fl^). Germline transmission black offspring were crossed with LysM-Cre transgenic mice (From the Jackson Laboratory, Strain #004781) to obtain myeloid cell-specific *Tm6sf2* knockout (mKO) mice (LysM Cre+/*Tm6sf2*^fl/fl^) that were then crossed with apolipoprotein E knockout (*ApoE*^−/−^) mice (from the Jackson Laboratory, Strain #002052) to generate LysM Cre+/*Tm6sf2*^fl/fl^*ApoE*^−/−^ (*Tm6sf2* mKO) and littermate LysM Cre−/*Tm6sf2*^fl/fl^/*ApoE*^−/−^ mice (Control) mice for the current study. In the atherosclerosis experiment, both male and female *Tm6sf2* mKO and littermate control mice were placed on a Western diet (WD) for 12 weeks (TD. 88137, ENVIGO, 42% of calories from fat and 0.2% cholesterol).

At the endpoint of experiment, animals were euthanized with CO_2_ overdose, and blood was collected for lipid profile at the Chemistry Laboratory of the Michigan Diabetes Research and Training Center (University of Michigan). Mice were perfused via the left cardiac ventricle with saline followed by 10% formalin. The whole aorta tree from the ascending aorta to the bifurcation of the common iliac arteries was isolated and stained with Oil Red O [15,16]. Sections from aortic roots (5 µm) were used for the quantification of plaque area, necrotic core, and macrophage content (Mac2) by a person who was blinded to the genotypes [15,16].

### 2.2. Oil Red O Staining

Whole aorta trees were isolated and stained with Oil Red O (#1024190250, Merck KGaA, Darmstadt, Germany), and en face lesion area was measured using National Institutes of Health (NIH) Image J software (ImageJ 1.53K, NIH, Bethesda, MD, USA, http://imagej.nih.gov/ij/, accessed on 6 July 2021) [15,16]. Sections and cells were stained according to the instructions from the manufacturer using microscopy (OLYMPUS cellSens Dimmension 1.18, Tokyo, Japan) [16].

### 2.3. THP-1-Derived Macrophage

THP-1 cells, a monocyte leukemia cell line, were purchased from the American Type Culture Collection (ATCC) (ATCC, Cat# TIB-202, 10801 University Blvd, Manassas, VA, USA) and kept in RPMI Medium 1640 (Gibco, 11875-093, New York, NY, USA) with 10% FBS, 100 U/mL penicillin, 100 µg/mL streptomycin, and 55 µM 2-mercaptoethanol (Gibco, 21985-023, 347 5th Ave, New York, NY, USA). Short-tandem repeat (STR) profiling of the THP-1 cell line was performed by ATCC. THP-1-derived macrophages were differentiated in the medium containing 100 ng/mL phorbol 12-myristate 13-acetate (PMA) (Sigma-Aldrich, Inc., P1585, St. Louis, MO, USA) for 72 h. After differentiation, macrophages were cultured in a fresh medium without PMA for at least 72 h before further experiments. THP-1-derived macrophages were transfected with non-Targeting siRNA (NT siRNA) or siTM6SF2 (TM6SF2 siRNA Horizon Discovery, Cat#MQ-032313-01-0005, Burwell, Cambridge, UK) at 20 nM for 48 h.

### 2.4. Isolation of Bone Marrow-Derived Macrophage (BMDM)

BMDMs isolated from *Tm6sf2* mKO and littermate control mice were cultured in bone marrow macrophage differentiation media [15]. The medium was based on Iscove’s Modified Dulbecco’s Medium (IMDM, #12440053, Gibco, New York, NY, USA) adding 25 ng/mL recombinant M-CSF (#416-ML-010, R&D Systems, Inc., Minneapolis, MN, USA), 10% FBS, 1% non-essential amino acids (#11140-050, Gibco, New York, NY, USA), 1 mM Sodium pyruvate (#11360-070, Gibco, New York, NY, USA), 50 pM-Mercaptoethanol (#31350010, Gibco, New York, NY, USA) and 100 U/mL penicillin, 100 µg/mL streptomycin (#15140122, Gibco, New York, NY, USA). Isolated bone marrow cells were induced for differentiation in IMDM containing M-CSF for seven days.

### 2.5. RNA Sequencing and Data Analysis

BMDMs were isolated from the control and mKO mice after a 12-week WD challenge. After differentiation, macrophages were treated with 100 μg/mL oxLDL (360-31, by Lowry protein assay, Lee Biosolutions, Inc., Maryland Heights, MO, USA) for 48 h. Each sample was collected from an individual animal (*n* = 4 for each group). RNA sample preparation by using the Qiagen RNeasy^®^ mini kit (#74106, Qiagen, Houston, TX, USA) and by following the manufacturer’s instructions. Samples were then sent to the Advanced Genomics Core of U-M Biomedical Research Core Facilities for next-generation sequencing and quality control analysis. The data analysis follows a previously published paper [17].

DESeq2 package (1.32.0) was used for differential gene expression analysis in R version 4.2.0. Genes with adjusted *p* values (*p-adj*) < 0.05 were considered significantly. For the signaling pathway and functional enrichment analysis of the differentially expressed genes (DEGs), KEGG enrichment, and gene ontology (GO) enrichment, Metascape was used for this study [18]. The heatmaps were provided by heatmap package (Raivo Kolde, version 1.0.12, View on GitHub) in R.

Gene set enrichment analysis (GSEA) [19] was used for determining the statistical significance between control and mKO samples, and figures were generated based on the differential expression rank. GSEA version 4.2.3 was used for the data analysis and followed the GSEA user guide. *p* < 0.05 were considered significant.

### 2.6. Cholesterol Content Measurement

THP-1-derived macrophages and BMDMs were treated with 100 μg/mL oxLDL for 24 h. After washing with PBS, cells were harvested for cholesterol/lipid extraction. Samples were resuspended by 200 μL Chloroform: Isopropanol: NP-40 (7:11:0.1) mix solution, spun at 15,000× *g* for 10 min, then all liquid was transferred to EP tube to dry at 37 °C overnight. Cholesterol was measured using a cholesterol fluorometric assay kit (Cayman Chemical, #10007640, Ann Arbor, MI, USA).

### 2.7. Flowcytometry

THP-1-derived macrophages were infected with non-targeting siRNA (NT siRNA) or siTM6SF2 (TM6SF2 siRNA) at 20 nM for 48 h. Then cells were loaded with 5 μg/mL Dil-oxLDL (#770262-9, Kalen Biomedical, Montgomery Village, MD, USA) at 37 °C for 40 min and then collected for flow cytometry to determine cholesterol uptake by Dil intensity. Data were analyzed using FlowJo_v10.8.1 at UMICH Flow Cytometry Core. Blank samples without Dil fluorescence were used as negative controls.

### 2.8. Real-Time Quantitative PCR Quantifies

Total RNA was extracted using the Qiagen RNeasy Mini Kit (Qiagen, #74106, Houston, TX, USA). Reverse transcription was performed using SuperScript™ III Reverse Transcriptase (Invitrogen, #18080044, Burlington, CA, USA). Real-time quantitative PCR (qRT-PCR) was performed using the Bio-Rad CFX-Time system and 2X Universal SYBR Green fast qPCR Mix (ABclonal, RM21203, Woburn, MD, USA) [15,16]. The qRT-PCR data were analyzed using the 2^−ΔΔCT^ method normalized to beta-actin (ACTB). Primers used in this study were listed in Appendix A.

### 2.9. Western Blot

Cellular proteins were extracted in RIPA lysis buffer supplemented with PhosSTOP™ and protease inhibitor cocktail (Roche, #04906837001, Mannheim, Germany). Protein extracts were resolved using BIO-RAD Mini-PROTEAN TGX precast Gels. Blotting membranes were incubated with primary antibodies (p-NFκB p65, Cell Signaling, #3033; NFκB p65, Cell Signaling, #4764; β-Actin, Cell Signaling Technology, #8H10D10, Danvers, MA, USA) at 4 °C overnight, washed then incubated with secondary antibodies (IRDye^®^ 680RD Donkey anti-Mouse, IRDye^®^ 800CW Donkey anti-Rabbit, LI-COR Biosciences, Lincoln, NE, USA) for 1 h. Images were captured and quantified using Image Studio Ver 3.1 CLx (LI-COR Biosciences, Lincoln, NE, USA).

### 2.10. Statistics

The data were analyzed using GraphPad Prism 9 (GraphPad Software, San Diego, CA, USA). RNA seq data were analyzed with R 4.1.3 (https://www.r-project.org/, accessed on 10 March 2022). A two-tailed unpaired Student’s t-test was used to compare two independent groups. One-way ANOVA or two-way ANOVA was used to compare more than two groups. Individual values were presented as interleaved scatter with bars plot mean ± SEM. *p* < 0.05 was considered statistically significant.

## 3. Results

### 3.1. Tm6sf2 Knockout in Myeloid Cells Inhibits Atherosclerosis in Mice

To study tissue-specific *TM6SF2* functions, we generated *Tm6sf2*^fl/fl^ mice that contained loxP sites flanking exons 2 of *Tm6sf2* (Appendix A). To investigate the effects of macrophage-specific *Tm6sf2* deficiency on atherosclerosis, we obtained myeloid-specific *Tm6sf2*-deficient mice on *ApoE*^−/−^ background by crossbreeding with mice expressing the LysM Cre transgene (Figure 1A,B). *Tm6sf2* depletion was confirmed in BMDMs from *Tm6sf2* mKO mice (LysM Cre+/*Tm6sf2*^fl/fl^
*ApoE*^−/−^) and littermate controls (LysM Cre−/*Tm6sf2*^fl/fl^
*ApoE*^−/−^) (Figure 1C). After 12 weeks of WD challenge, male myeloid-specific *Tm6sf2* deficient mice significantly slowed down the development of atherosclerosis compared to littermate control, as evidenced by the reduced plaque size in whole aortas by *en face* Oil Red O staining (Figure 1D). Frozen sections of aortic roots were subjected to hematoxylin and eosin (H&E) staining, Oil Red O staining, and macrophage immunofluorescent staining using an antibody against Mac2/galectin 3, a widely used macrophage marker in mouse studies [16]. Compared to littermate control mice, aortic sinus sections from *Tm6sf2* mKO mice demonstrated significantly decreased plaque and necrotic core areas (Figure 1E). Oil Red O staining showed less lipid accumulation in the atherosclerotic plaques in *Tm6sf2* mKO mice (Figure 1F). Along with the decreased lipid accumulation, the Mac2-positive area in atherosclerotic plaques in *Tm6sf2* mKO mice was significantly lower than that in littermate controls (Figure 1G). There were no significant differences in body weight (Figure 1H), liver weight/body weight ratio (Figure 1J), blood glucose (Appendix A), and insulin (Appendix A) levels between control and mKO mice. Myeloid-specific *Tm6sf2* knockout did not affect serum TG, TC, LDL-C, and HDL- levels (Figure 1J). Samiliar results were found in female mKO and control mice (Appendix A). These in vivo results indicate that myeloid *Tm6sf2* knockout can inhibit atherosclerosis development without changing the lipid metabolism and glucose metabolism.

### 3.2. Differential Expression Analysis of BMDMs Isolated from Control and Tm6sf2 mKO Mice

Atherogenesis is a complex disease, and many factors contribute to the progression of atherogenesis. Cholesterol metabolism and local inflammation are two important factors for forming atherosclerotic plaques [20]. Macrophages have been considered a fundamental element in the development of atherosclerosis [21]. To unbiasedly understand the mechanisms by which myeloid-specific *TM6SF2* contributes to atherosclerosis development, BMDMs from control and Tm6sf2 mKO mice were used for whole-transcriptome analysis (RNA-Seq). After the analyses, more than 24 million reads were generated from each sample. Compared with the control group, there were 3681 genes downregulated and 5487 genes upregulated (*padj* < 0.05) in the mKO group, shown in the heatmap (Figure 2A). Next, we used GO and KEGG enrichment to analyze the involved signaling pathways. Among the top 20 significant overrepresented pathways (Figure 2B), inflammation-associated pathways were the most prominent, such as regulation of cytokine production (GO:0001817), inflammatory response (GO:0006954), cytokine-mediated signaling pathway (GO:0019221), and cytokine signaling in the immune system (R-MMU-1280215). As GSEA (Gene Set Enrichment Analysis) shows (Figure 2C,D), 163 genes in HALLMARK_INFLAMMATORY_RESPONSE pathway were positively correlated with control, and 31 genes were negatively correlated with mKO (Figure 2C). All 24 genes in REACTOME_CHOLESTEROL_BIOSYNTHESIS pathway were positively correlated with the control group (Figure 2D), consistent with the findings using GO enrichment analysis (data showed in Appendix A). In the HALLMARK_TNFA_SIGNALING_VIA_NFKB pathway, 137 genes were positively correlated with control and 58 genes were negatively correlated with mKO (Figure 2E). Furthermore, we found that the *Hspa5* (*Bip*), *Xbp1*, and *Nfkb1* (NF-κB) genes were all down-regulated in Tm6sf2 KO BMDMs (Figure 2F). The inflammatory genes such as *Il6*, *Il1b*, and *Ccl2* (*Mcp-1*) were also down-regulated in mKO BMDMs. The Bip-IRE-1α-Xbp1 signaling pathway regulates ER stress [22], which is associated with various pathological changes. This pathway also activates the NF-κB pathway, a classical inflammation transcriptional signaling pathway [23,24]. Our data suggest that *Tm6sf2* deficiency reduces the inflammatory responses in BMDMs and may lead to the alleviation of atherosclerosis in mice.

### 3.3. TM6SF2 Regulates Inflammatory Responses in Macrophages

We then determined whether macrophage *TM6SF2* contributes to inflammatory responses in pro-atherosclerotic conditions. Ox-LDL, a well-established atherosclerosis risk factor, was used to induce inflammatory responses in human and mouse macrophages. oxLDL treatment upregulated the *TM6SF2* expression in THP-1-derived macrophages and in BMDMs (Appendix A). THP-1-derived macrophages were transfected with siTM6SF2 (TM6SF2 siRNA) or non-Targeting siRNA (NT siRNA), and the knockdown efficiency was confirmed as shown in Appendix A. We found that knockdown of *TM6SF2* abolished oxLDL-induced inflammatory responses, indicated by the reduced expressions of TNF, IL1B, and CCR2 (Figure 3A), while the overexpression of *TM6SF2* dramatically enhanced oxLDL-induced inflammatory responses in THP-1-derived macrophages (Figure 3B). Using BMDMs isolated from control and mKO mice, we confirmed that deficiency of *Tm6sf2* showed reduced inflammatory responses (Figure 3C). We further demonstrated that the overexpression of *TM6SF2* in wild-type BMDMs exacerbated ox-LDL-induced inflammation (Figure 3D). These findings indicate that *TM6SF2* plays a role in the inflammatory responses in macrophages.

### 3.4. TM6SF2 Contributes to Foam Cell Formation

Our RNA-seq data suggest that *TM6SF2* plays a role in cholesterol handling in macrophages. oxLDL induces the transformation of THP-1-derived macrophages and BMDMs to cholesterol-rich foam cells, evidenced by cholesterol uptake assay, Oil Red O staining, and intracellular cholesterol contents measurement (Figure 4). Knockdown of *TM6SF2* in THP-1-derived macrophages significantly reduced cholesterol uptake and foam cell formation (Figure 4A,B), while the overexpression of *TM6SF2* facilitated oxLDL-induced cholesterol uptake and foam cell formation (Figure 4C,D). Similar findings were confirmed in BMDMs isolated from control and Tm6sf2 mKO mice, as well as in *TM6SF2* overexpressing BMDMs (Figure 4E–H). These data suggest that *TM6SF2* contributes to macrophage cholesterol handling and foam cell formation.

### 3.5. TM6SF2 Positively Regulates ER Stress Pathway in Macrophages

As *TM6SF2* is an ER membrane protein, our previous study also demonstrated that *TM6SF2* interacts with the Bip-associated ER stress sensor inositol-requiring enzyme (*IRE1α*) and affects its downstream *XBP1* in the liver [25]. In macrophages, our RNA-seq data also indicated that *TM6SF2* modulated the ER stress pathway. oxLDL induced ER stress in macrophages (Figure 5). We showed that the overexpression of *TM6SF2* in THP-1-derived macrophages and BMDMs from C57BL/6J mice upregulated the expression of ER stress markers such as Bip (*Hspa5*), IRE1α (*Ern1*), JNK (*Mapk8*), ASK1 (*Map3k5*), and XBP1 (*Xbp1*) (Figure 5B,D). However, the expression of these ER stress markers was downregulated in *TM6SF2*-deficient macrophages (Figure 5A,C), indicating that *TM6SF2* positively regulates ER stress pathway in macrophages.

### 3.6. TM6SF2 Activates NF-κB in Macrophages

NF-κB pathway can be activated by many atherosclerotic risk factors as well as ER stress, intensifying inflammatory gene expression and inflammation cascade. To further investigate the effects of *TM6SF2* on inflammatory responses and ER stress through regulating the NF-κB signaling pathway in macrophages, we determined the phosphorylation of p65 at S536, which contributes to nuclear translocation and the induction of target gene expression. As shown in Figure 6, oxLDL induced phosphorylation of p65. Knockdown of *TM6SF2* abolished the effect of oxLDL on p65 activation as well as suppressed the expression of p65 (Figure 6A), indicating that TM6SF2 contributes to the activation of NF-κB pathway in macrophages. Indeed, the overexpression of TM6SF2 enhanced the activation of p65 in THP-1-derived macrophages (Figure 6B), consistent with the findings of *TM6SF2* in inflammatory responses and ER stress.

## 4. Discussion

Exome-wide association studies and genome-wide association studies have identified that the *TM6SF2* variant (encoding p. Glu167Lys, E167K) is associated with decreased myocardial infarction risk and increased fatty liver disease [26,27]. Here, our data demonstrate that myeloid cell-specific *Tm6sf2* deficiency inhibits atherosclerosis development independent of the circulating lipid levels. Regarding the mechanism, we discovered that TM6SF2 contributes to inflammatory responses, foam cell formation, and ER stress in macrophages. Therefore, this study determined the role of the macrophage TM6SF2 in atherosclerosis and put forward a new target for treating this disease.

Cardiovascular disease is one of the leading causes of death worldwide [28,29]. The success of genetic studies has significantly advanced our understanding of the etiology of ASCVD and provided new therapeutic strategies, particularly via identifying drug targets for ASCVD treatment. One typical representative is proprotein convertase subtilisin/kexin type 9 serine protease (PCSK9), emerging as an effective cholesterol-lowering target [30,31]. Human genetic studies have documented that the *TM6SF2* variant E167K is associated with decreased TC and TG levels and reduced cardiovascular arterial disease risk [7,8,9], presenting cardiovascular protective properties. However, the same variant in *TM6SF2* is also associated with increased risks of nonalcoholic fatty liver disease (NAFLD) and diabetes. The E167K mutation in *TM6SF2* yields decreased protein levels due to increased degradation [7]. The overexpression of loss-of-function TM6SF2 reduces plasma lipid levels and atherosclerosis development [32]. Knockdown or inactivation of TM6SF2 attenuates apolipoprotein B lipidation and VLDL secretion from hepatocytes, decreasing circulating TC and TG levels while causing lipid accumulation in the liver [13,32,33]. The effects of TM6SF2 in the liver are confirmed in TM6SF2-E167K knock-in mice [25]. Liver-specific *Tm6sf2*-deficient mice spontaneously exhibit hepatic steatosis, fibrosis, and accelerated development of hepatocellular cancer [11,13,25]. Therefore, the increased NAFLD risk by TM6SF2 inactivation limits its application for cardiovascular arterial disease treatment via regulating lipid metabolism in the liver. Unlike the liver-specific or systemic studies, our data demonstrate that myeloid-specific *Tm6sf2* deficiency attenuates atherosclerosis development independent of circulating lipid levels. As expected, we did not observe any changes in body weight, blood glucose, liver weight, or body weight, indicating that myeloid cell *Tm6sf2* deficiency does not affect the lipid and glucose metabolism in the liver.

Lipoproteins accumulated in the subendothelial area recruit monocyte/macrophages, T-cells, mast cells, and other immune cells to the lesion sites, where monocyte-differentiated macrophages take up cholesterol and become foam cells, further promoting atherosclerotic plaque progression [21,34,35]. We observed reduced atherosclerotic plaque area and Mac2-positive area in myeloid cell-specific *Tm6sf2*-deficient mice, suggesting that a deficiency of *Tm6sf2* prevents macrophage infiltration into the aortic wall. Vascular chronic inflammation further recruits monocytes and macrophages into atherosclerotic lesions. Macrophages uptake cholesterol to form foam cells, which is the key element during the plaque formation and progression of atherosclerosis [4,36]. Our data show that *Tm6sf2*-deficiency affects the inflammation-associated signaling pathways, such as via the regulation of cytokine production, inflammatory response, and cytokine-mediated signaling pathway, which are all involved in macrophage movement and infiltration. oxLDL, a major risk factor for the initiation and development of atherosclerosis, upregulated TM6SF2 expression in macrophages and increased TM6SF2 expression enhances oxLDL-induced inflammatory responses and foam cell formation, showing a vicious circle of positive feedback. TM6SF2-deficiency blocks this feedback and reduces inflammatory responses and foam cell formation in macrophages, presenting potential protective mechanisms underlying the alleviation of atherosclerosis in *Tm6sf2* mKO mice. In endothelial cells, disturbed flow activates crucial proatherogenic pathways [37]. Single-cell RNA sequencing and scATACseq study have demonstrated that TM6SF2 expression is upregulated during disturbed flow-induced endothelial cell phenotype change from atheroprotective phenotypes to proinflammatory cells [38]. These results suggest that TM6SF2 in non-hepatocytes, such as macrophages and endothelial cells, plays a role in atherosclerosis development.

In atherosclerotic lesions, macrophages are the prominent cells undergoing ER stress. Lipoprotein and inflammatory cytokines induce ER stress in macrophages and foam cells, causing apoptosis and recruiting more macrophages to the atherosclerotic sites to accelerate progression of atherosclerosis [39]. TM6SF2 is localized in the ER membrane and ERGIC (ER-Golgi intermediate compartment) [40]. TM6SF2 deletion or loss-of-function mutation affects ER ultrastructure and induces ER stress markers in zebrafish and human livers due to lipid accumulation under physiological conditions [33]. However, loss-of-function of TM6SF2 in E167K knock-in mice are resistant to high-fat-diet-induced ER stress in the liver [25]. Consistent with the findings from the high-fat-diet challenge study [25], after a 12-week WD challenge, the expression of ER stress markers are downregulated in BMDMs from *Tm6sf2* mKO mice compared with that from littermate controls. We also confirmed that oxLDL upregulates the expression of ER stress markers, which can be enhanced by overexpression of TM6SF2 in macrophages, while TM6SF2 deficiency largely abolishes its effects, consistent with the protective effects during atherosclerosis development. These findings show that TM6SF2 may function differently in regulating ER stress under adaptive processes and pathological conditions.

In response to pro-atherosclerotic and stress stimuli, NF-κB members are activated, and the NF-κB signaling pathway is vital for inflammatory responses and ER stress [23,41]. ER stress also activates the NF-κB signaling pathway and causes the translocation to the nucleus to activate pro-inflammatory gene expression [24,42,43]. Our transcriptome analysis revealed that TM6SF2 contributes to the activation of the NF-κB signaling pathway in macrophages. The phosphorylation of Ser536 in the cytosolic p65 promotes its nuclear translocation and facilitates p65 binding to specific promoter sequences [44]. We found that oxLDL activates NF-κB, reflected in the increased phosphorylation of the NF-κB p65 subunit, in a TM6SF2-dependent manner. Furthermore, we found that knockdown of TM6SF2 decreased the total p65 level in macrophages. As TM6SF2 is an ER membrane protein, it may regulate the p65 level through posttranslational modifications, such as degradation, ubiquitination, or acetylation. Further investigation of the role of TM6SF2 in p65 expression and activation is warranted in our follow-up studies.

It should be noted that oxLDL induces the expression of TM6SF2 in macrophages. Further examination of the intracellular mechanisms underlying the regulation of TM6SF2 expression will be necessary for clarifying the function of TM6SF2 in atherosclerosis development fully. Whether TM6SF2 in other vascular cells, such as endothelial cells and vascular smooth muscle cells, plays a role in atherosclerosis is unknown. Our future studies will explore the functions of TM6SF2 in atherosclerosis using endothelial cell or vascular smooth muscle-specific *Tm6sf2* knockout mouse models, as well as loss of function of TM6SF2 in E167K knock-in mice.

## 5. Conclusions

In conclusion, our findings demonstrate that myeloid cell-specific *Tm6sf2* deficiency inhibits atherosclerosis development in mice without affecting lipid and glucose metabolism. We discovered that a deficiency of TM6SF2 in macrophages inhibits inflammatory responses, foam cell formation, and ER stress upon oxLDL stimulation. Therefore, the repression of TM6SF2 in macrophages could be a new therapeutic target for atherosclerosis treatment.

## Figures and Tables

**Figure 1 cells-11-02877-f001:**
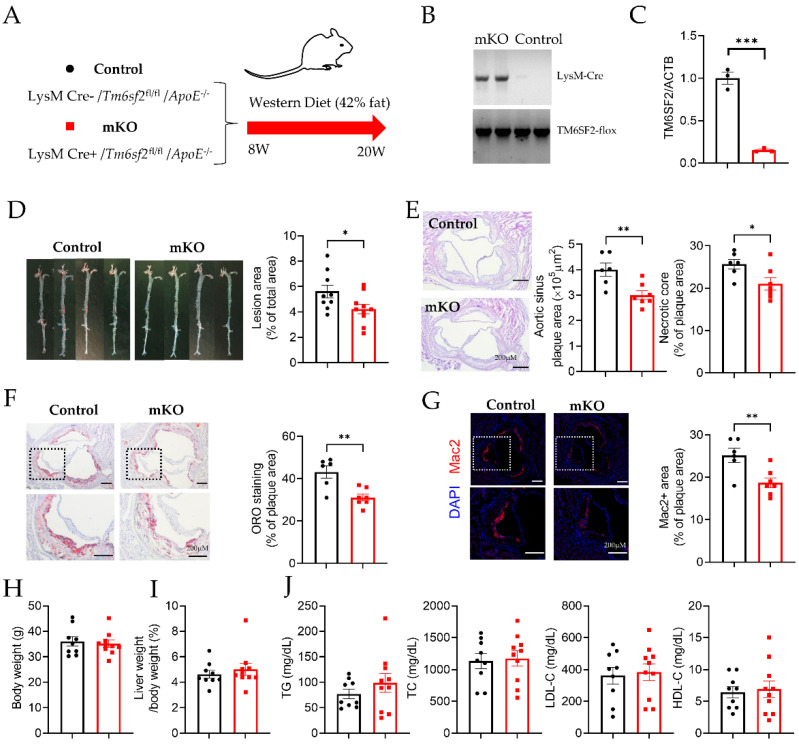
*Tm6sf2* Knockout in Myeloid Cells Inhibits Atherosclerosis Development in Mice. (**A**) Schematic of transgenic animal model for atherosclerosis study. Male control (LysM Cre−/*Tm6sf2*^fl/fl/^*ApoE*^−/−^) and mKO (LysM Cre+/*Tm6sf2*^fl/fl/^*ApoE*^−/−^) mice were fed a Western diet for 12 weeks start from age at 8 weeks. (**B**) Representative genotyping results from Control and mKO mice. (**C**) Real-time PCR analysis of *Tm6sf2* expression (normalized to *Actb*) in BMDMs from control (black) and mKO (red) mice, *n* = 3. (**D**) The aortas from control and mKO mice were dissected, stained with Oil Red O, split longitudinally, and pinned open for surface lesion measurements. The lesion area is quantified using Image J. *n* = 9 for control, *n* = 10 for mKO. E-G, Representative aortic sinus tissue-section images H&E staining (**E**), ORO staining (**F**), and Mac-2 immunofluorescent staining (**G**). Scale bar = 200 µm. *n* = 6 for control, *n* = 7 for mKO. (**H**) Bodyweight and (**I**) liver weight/body weight ratio, (**J**) Circulating TG, TC, LDL-C, and HDL-C levels, *n* = 9 for control, *n* = 10 for mKO. All data are presented as mean ± SEM, * *p* < 0.05, ** *p* < 0.01, *** *p* < 0.001 as indicated.

**Figure 2 cells-11-02877-f002:**
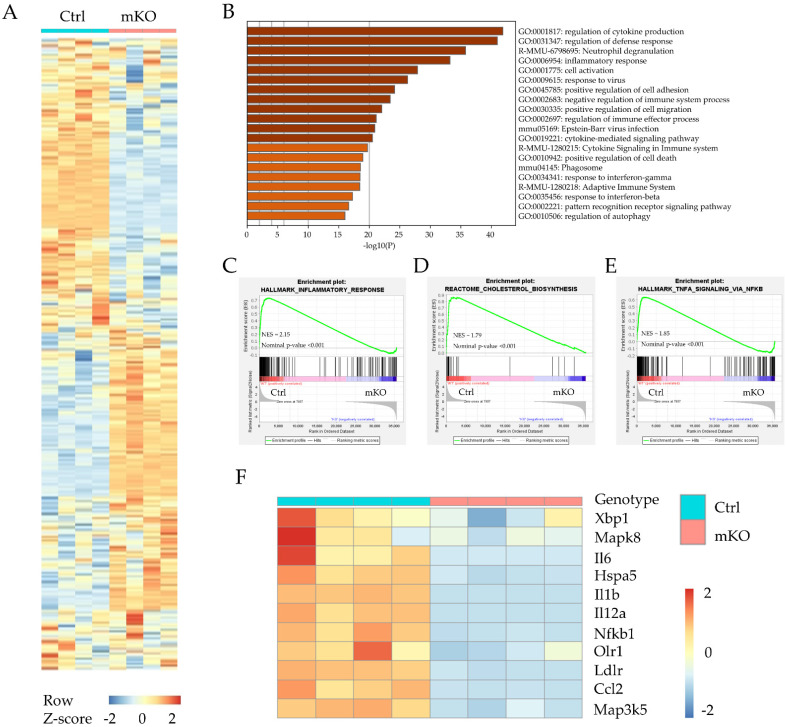
RNA sequencing differential expression analysis between BMDMs from Control (Ctrl) and *Tm6sf2* mKO (mKO) mice. (**A**) Heatmap representation depicting the significantly changed genes in BMDMs after oxLDL treatment (*Tm6sf2* mKO compared with the control, *padj* < 0.05). (**B**) Visualizations of GO enrichment of the significantly changed genes by metascape. Here are the top 20 significant signaling pathways. (**C**–**E**) Details of selected signaling pathway enrichment plot demonstrated by GSEA analysis, which shows the specific gene distribution and correlation with control and mKO BMDMs. (**F**) Heat map representation depicting the expression of genes associated with inflammatory, ER stress, and cholesterol transport in BMDMs upon oxLDL stimulation (*Tm6sf2* mKO compared with the control. *p* ≤ 0.01 and log2-fold change ≥0.5, the color bar indicates z-score in heatmap).

**Figure 3 cells-11-02877-f003:**
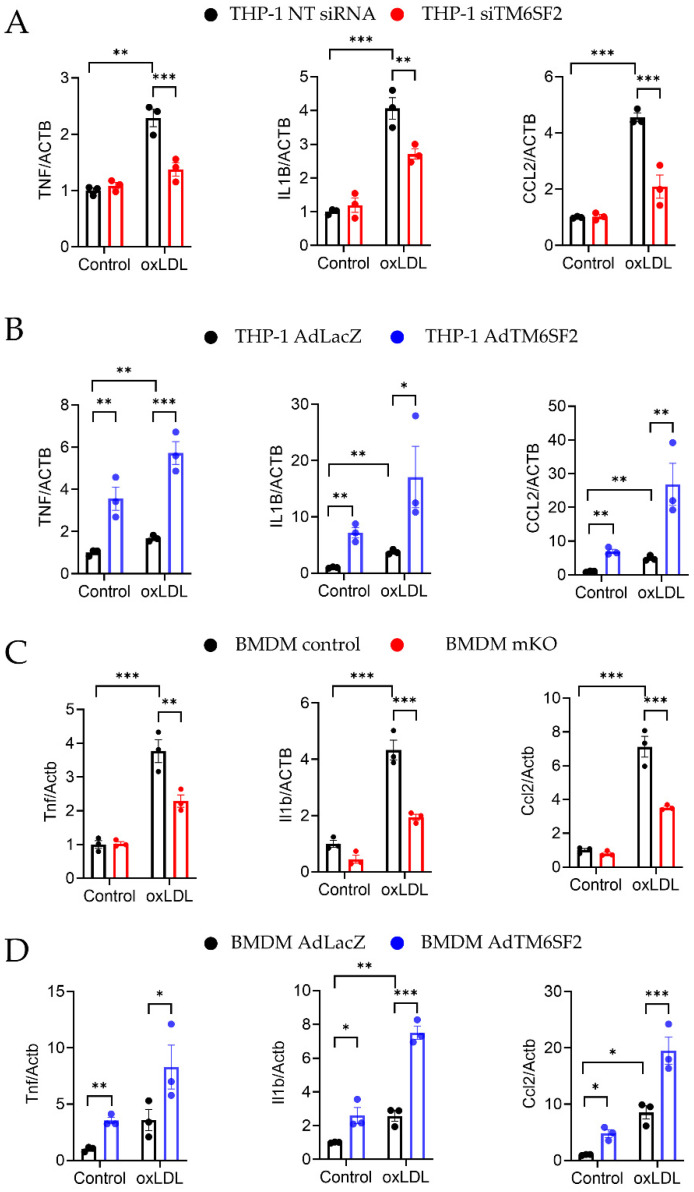
*TM6SF2* Contributes to Inflammatory Responses in Macrophages. (**A**) THP-1-derived macrophages were transfected with TM6SF2 siRNA or NT siRNA (20 nM) for 48 h and treated with oxLDL (100 µg/mL) for 4 h. (**B**) THP-1-derived macrophages were transfected with AdLacZ or AdTM6SF2 (100 MOI) for 48 h and treated with oxLDL (100 µg/mL) for 4 h. (**C**) BMDMs were isolated from control and Tm6sf2 mKO mice and treated with oxLDL (100 µg/mL) for 4 h. (**D**) BMDMs were isolated from wild type C57BL/6J mice and were transfected with AdLacZ or AdTM6SF2 (100 MOI) for 48 h and treated with oxLDL (100 µg/mL) for 4 h. mRNA expression of *TNF-α*, *IL1β*, and *CCL2* were measured by qRT-PCR. *n* = 3 for each group. All data are presented as mean ± SEM, statistics by 2-way ANOVA. * *p* < 0.05, ** *p* < 0.01, *** *p* < 0.001 as indicated.

**Figure 4 cells-11-02877-f004:**
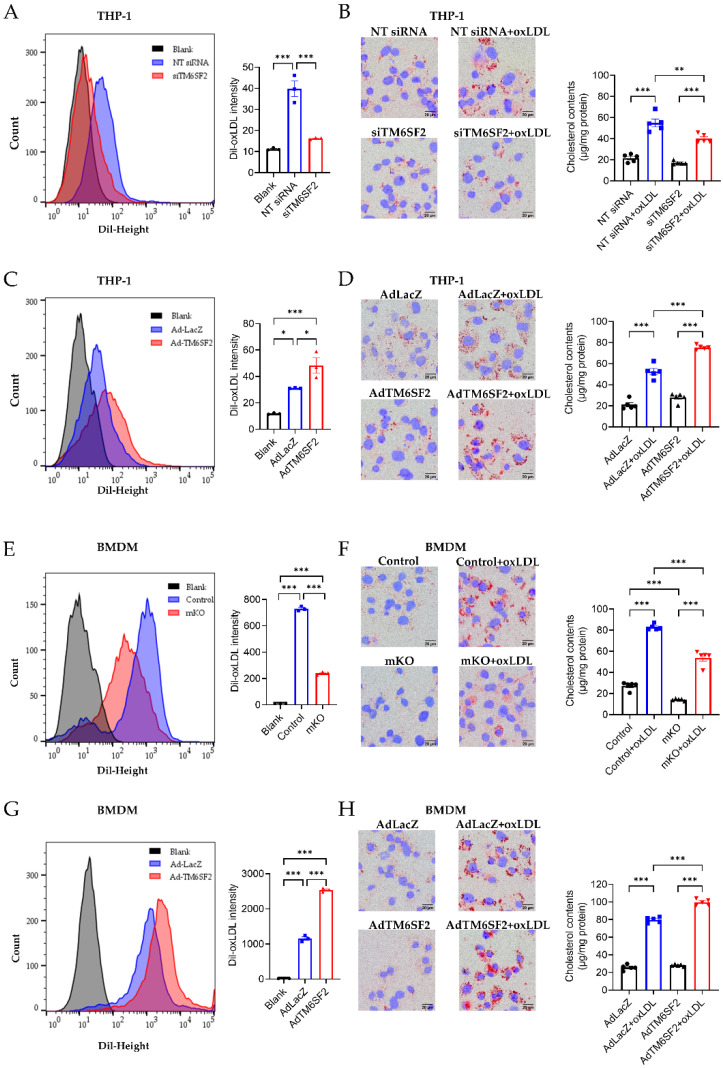
*TM6SF2* contributes to foam cell formation. Macrophages were loaded with 5 μg/mL Dil-oxLDL at 37 °C for 40 min and collected for flow cytometry to detect the cholesterol uptake. Macrophages were loaded with 100 μg/mL oxLDL overnight at 37 °C and foam cell formation was determined by Oil Red O staining and the cellular cholesterol content (normalized to total protein). (**A**,**B**) THP-1-derived macrophages were transfected with NT siRNA or TM6SF2 siRNA (20 nM) for 48 h and treated with oxLDL (100 µg/mL) overnight. (**C**,**D**) THP-1-derived macrophages were transfected with AdLacZ or AdTM6SF2 (100 MOI) for 48 h and treated with oxLDL (100 µg/mL) overnight. (**E**,**F**) BMDMs were isolated from control and mKO mice and treated with oxLDL (100 µg/mL) overnight. (**G**,**H**) BMDMs were isolated from wild-type C57BL/6J mice and were transfected with AdLacZ or AdTM6SF2 (100 MOI) for 48 h and treated with oxLDL (100 µg/mL) overnight. *n* = 3 for flow cytometry, *n* = 5 for cholesterol contents detection. All data are presented as mean ± SEM, statistics by ordinary one-way ANOVA. * *p* < 0.05, ** *p* < 0.01, *** *p* < 0.001 as indicated.

**Figure 5 cells-11-02877-f005:**
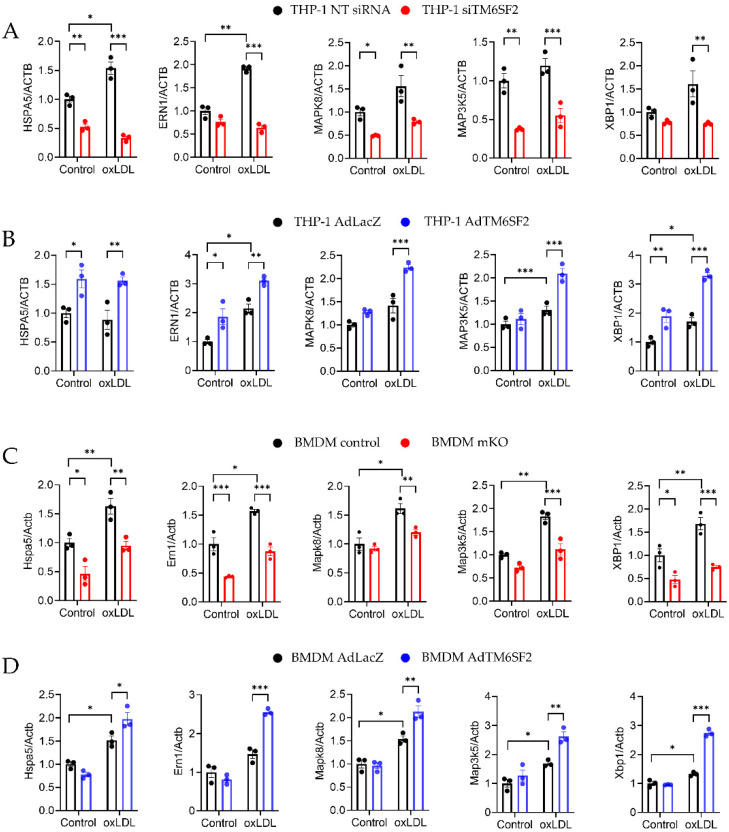
*TM6SF2* Regulates ER stress-related Gene Expression in Macrophages. Macrophages were loaded with 100 µg/mL oxLDL overnight at 37 °C and mRNA expression of *BIP*, *IRE1α*, *JNK*, *ASK1* and *XBP1* were measured by qRT-PCR. (**A**) THP-1-derived macrophages were transfected with TM6SF2 siRNA or NT siRNA (20 nM) for 48 h and (**B**) THP-1-derived macrophages were infected with AdTM6SF2 or AdLacZ (100 MOI) for 48 h. (**C**) BMDMs were isolated from control and mKO mice and treated with oxLDL (100 µg/mL) overnight. (**D**) BMDMs were isolated from wild type C57BL/6J mice and were transfected with AdLacZ or AdTM6SF2 (100 MOI) for 48 h and treated with oxLDL (100 µg/mL) overnight. *n* = 3 for each group. All data are presented as mean ± SEM, statistics by 2way ANOVA. * *p* < 0.05, ** *p* < 0.01, *** *p* < 0.001 as indicated.

**Figure 6 cells-11-02877-f006:**
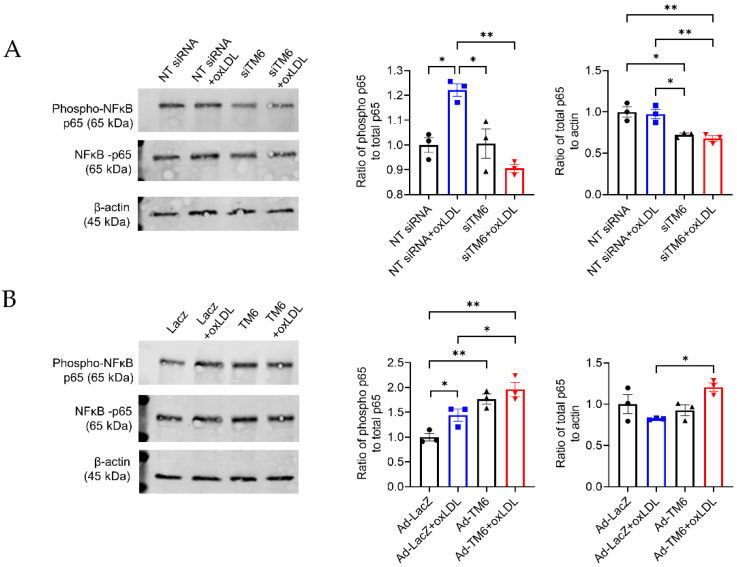
*TM6SF2* increases p65 phosphorylation in THP-1-derived macrophages. (**A**) THP-1-derived macrophages were transfected with NT siRNA or TM6SF2 siRNA (20 nM) for 48 h and treated with oxLDL (100 µg/mL) for 1 h. (**B**) THP-1-derived macrophages were transfected with AdLacZ or AdTM6SF2 (100 MOI) for 48 h and treated with oxLDL (100 µg/mL) for 1 h. phospho-p65 and total p65 levels were determined by Western Blotting. Representative results and quantifications were shown. *n* = 3 for each group. All data are presented as mean ± SEM, statistics by Ordinary one-way ANOVA. * *p* < 0.05, ** *p* < 0.01 as indicated.

## Data Availability

Not applicable.

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
