# Peer review of "Myeloid TM6SF2 Deficiency Inhibits Atherosclerosis"

_cells, 2022, doi:10.3390/cells11182877_

Round 1

Reviewer 1 Report

In this paper, Zhu et al. examined a potential role of myeloid TM6SF2 in atherosclerosis. Genetic studies showed an inverse correlation of certain TM6SF2 variant with plasma LDL-C and ASCVD. However, a potential role of TM6SF2 in the function of myeloid cells, particularly macrophages, and consequently atherosclerosis has never been reported. Using mouse models of atherosclerosis, the authors first demonstrated that ablation of TM6SF2 in myeloid cells reduced atherosclerosis without changing plasma lipid levels. Further studies by RNA-sequencing of BMDMs from TM6 mKO mice revealed downregulation of genes associated with inflammation, cholesterol uptake, and ER stress. Then by using THP1 macrophages and mouse BMDMs, the authors demonstrated that oxLDL treatment upregulated TM6SF2 in macrophages. Silencing TM6SF2 in THP1 macrophages or Tm6sf2 deficiency in BMDMs reduced inflammatory responses and ER stress and attenuated cholesterol uptake and foam cell formation, while overexpression of TM6SF2 in these cells had opposite effects in response to oxLDL treatment. Thus, the authors concluded that myeloid TM6SF2 deficiency inhibits atherosclerosis development and that TM6SF2 in myeloid cells is a potential therapeutic target for ASCVD prevention.

              This is a well-designed study combining mouse models of atherosclerosis and tissue cultures, in which both loss-of-function and gain-of-function approaches were performed. The study demonstrated a novel role of TM6SF2 in macrophage inflammation and foam cell formation and subsequently atherosclerosis. The data provided are strong and convincing. The paper was very well written. Some minor weaknesses were identified.

   1. The authors only provided the source of DiI-oxLDL, but not unlabeled oxLDL. Please provide the source of the unlabeled oxLDL. Did the concentration (100 ug/ml) of oxLDL used in tissue culture refer to protein or cholesterol?

   2. Please also provide the sex of mice used, were both males and females used or just one sex used? If one sex, the authors need to discuss this as a limitation of this study.

Author Response

Reviewer #1: This is a well-designed study combining mouse models of atherosclerosis and tissue cultures, in which both loss-of-function and gain-of-function approaches were performed. The study demonstrated a novel role of TM6SF2 in macrophage inflammation and foam cell formation and subsequently atherosclerosis. The data provided are strong and convincing. The paper was very well written. Some minor weaknesses were identified.

  1. The authors only provided the source of DiI-oxLDL, but not unlabeled oxLDL. Please provide the source of the unlabeled oxLDL. Did the concentration (100 ug/ml) of oxLDL used in tissue culture refer to protein or cholesterol?

A: We thank the reviewer for the critical questions. The oxLDL was purchased from LEE Biosolutions (Catalogue Number: 360-31), and the concentration of oxLDL was measured by Lowry protein assay.

  1. Please also provide the sex of mice used, were both males and females used or just one sex used? If one sex, the authors need to discuss this as a limitation of this study.

A: We appreciate the reviewer for this concern. We did atherosclerosis studies using both male and female Tm6sf2 knockout and littermate control mice. The results from female mice were provided in Figure S3 in this revised version. Although female Tm6sf2 knockout mice also showed decreased atherosclerotic size, we submitted the related data as a supplemental Figure due to the small animal number in each group.

Reviewer 2 Report

This study reports a myeloid specific knockout of a transmembrane protein (TM6SF2) in mice and effects due to high fat diet in inducing atherosclerosis-mediated effects. The study is well performed and the data is well organized and put together. The manuscript is also written clearly for the most part. Major and Minor changes are requested and is left to the discretion of authors and editors.

Major changes

1. The premise of the study is that TM6SF2 is an important target for atherosclerosis. The exome-wide and GWAS studies have identified a variant (E167K) in this gene. Thus, although gene knockout studies are valuable in gaining insight for the gene function in a particular cell type that is relevant to disease, in order for reaching translational relevance for this target, it would be nice to show the effects of TM6SF2 E167K protein in macrophages, at least in vitro. Demonstrating this fact will greatly enhance the clinical value and relevance of this paper. 

2. ORO quantification is needed in figure 4.

3. NF-kB western blot data (Fig. 6) could be supplemented with NF-kB reporter assay data to increase the rigor of the findings.

4. The findings linking TM6SF2 and NF-kB & inflammation are a bit associative. Can a direct link or a rescue experiment be done to make this "connection finding," a bit more relevant?

5. The Discussion section needs to have some more critical insight into the findings rather rehash the results section. An assessment of what their findings mean in relation to previous work, and cells associated with atherosclerosis disease progression and target TM6SF2 would be helpful.

Minor changes

1. Please provide a signaling cartoon to help capture your findings.

2. A reference is needed for Mac2 macrophage marker in mouse (Results section 3.1 - page 4).

3. Section 3.3 (page 7) - Line 4. I think it should be Figure S4 and Figure 3 in the brackets. Also, on line 8, Figure 3A - can you include your marker list here for easy reference.

4. Also, please check throughout the results section for marker inclusion in the relevant figure call outs. 

Author Response

Reviewer #2: This study reports a myeloid specific knockout of a transmembrane protein (TM6SF2) in mice and effects due to high fat diet in inducing atherosclerosis-mediated effects. The study is well performed and the data is well organized and put together. The manuscript is also written clearly for the most part. Major and Minor changes are requested and is left to the discretion of authors and editors.

Major changes

  1. The premise of the study is that TM6SF2 is an important target for atherosclerosis. The exome-wide and GWAS studies have identified a variant (E167K) in this gene. Thus, although gene knockout studies are valuable in gaining insight for the gene function in a particular cell type that is relevant to disease, in order for reaching translational relevance for this target, it would be nice to show the effects of TM6SF2 E167K protein in macrophages, at least in vitro. Demonstrating this fact will greatly enhance the clinical value and relevance of this paper.

A: We thank the reviewer for the good suggestion. The E167K mutation in TM6SF2 yields a misfolded protein, causing accelerated protein degradation and reduced protein levels in hepatocytes (PMID: 24978903). In the current manuscript, siRNA was used to knockdown TM6SF2 in macrophages to mimick the reduced TM6SF2 levels. we added one sentence to the limitation part: “Our future studies will explore the functions of TM6SF2 in atherosclerosis using endothelial cell or vascular smooth muscle-specific Tm6sf2 knockout mouse models, as well as loss-of-function of TM6SF2 in E167K knock-in mice.”  

  1. ORO quantification is needed in figure 4.

A: We thank the reviewer for the question. Oil Red O is a fat-soluble dye that stains neutral triglycerides and lipids. As the RNA sequencing findings suggest that TM6SF2 may contribute to cholesterol handling in macrophages, we measured the total introcellular cholesterol levels.

  1. NF-kB western blot data (Fig. 6) could be supplemented with NF-kB reporter assay data to increase the rigor of the findings. And 4. The findings linking TM6SF2 and NF-kB & inflammation are a bit associative. Can a direct link or a rescue experiment be done to make this "connection finding," a bit more relevant?

A: We thank the reviewer for the good suggestion. As TM6SF2 is an ER membrane protein, in the future studies, we will discuss how TM6SF2 regulates the expression and activation of NF-kB & inflammation pathways through transcriptional level, RNA stability, translation, and posttranslational modifications.

  1. The Discussion section needs to have some more critical insight into the findings rather rehash the results section. An assessment of what their findings mean in relation to previous work, and cells associated with atherosclerosis disease progression and target TM6SF2 would be helpful.

A: This concern is the hard part for us when we prepared this manuscript because there are so few non-hepatocyte studies published. We only found one publication using single-cell RNA sequencing and scATACseq analysis, in which TM6SF2 expression is upregulated during disturbed flow-induced endothelial cell phenotype change from atheroprotective phenotypes to proinflammatory cells (PMID: 33326796).

Minor changes

  1. Please provide a signaling cartoon to help capture your findings.

A: We thank the reviewer for the good suggestion. Currently we do not have a whole picture of the upstream and downstream signal pathway of TM6SF2. Our future research will fill in this pathway.

  1. A reference is needed for Mac2 macrophage marker in mouse (Results section 3.1 - page 4).

A: A reference is added.

  1. Section 3.3 (page 7) - Line 4. I think it should be Figure S4 and Figure 3 in the brackets. Also, on line 8, Figure 3A - can you include your marker list here for easy reference.

A: We thank the reviewer for the good suggestion. In preparing the manuscript, we debated whether to put this result into Figure 3, but the Figure pattern in Figure 3 was changed.

  1. Also, please check throughout the results section for marker inclusion in the relevant figure call outs.

A:We thank the reviewer for the kind reminder. We checked the Figure orders.